# Spatial and temporal characteristics of surface soil moisture in a disturbed coal mining area of Chinese Loess Plateau

**Yi Cao, Hongfen Zhu, Rutian Bi\*, Yaodong Jin**

College of Resource and Environment, Shanxi Agricultural University, Shanxi, China

\* birutian@163.com

**Data Availability Statement:** All relevant data are within the manuscript and its Supporting Information files.

**Funding:** This research was funded by the National Key Research and Development Program of China (2018YFD0200401), and the Natural Science

## Abstract

Soil water content is an important variable in hydrology and many related disciplines. It affects runoff from precipitation, groundwater recharge, and evapotranspiration. This research used the coal mining area of the Changhe River Basin in the Loess Plateau as a study and using SAR (Synthetic Aperture Radar) data, the surface soil water in 24 days (From Jan 25, 2018 to Dec 10, 2019) was estimated using a radar signal change detection algorithm. The temporal and spatial variation characteristics of surface soil water inside and outside the disturbed area were compared and analyzed. An empirical orthogonal function (EOF) analysis method was used to analyze the potential temporal and spatial variation of surface soil water, and to detect the regional soil water variation under coal mining disturbances to better understand the different potential modes of spatial variation of soil water in the unobserved time. The results showed that the average surface soil water content in the study area changed with season, showing a dry-wet-dry variation. Moreover, it was significantly affected by precipitation factors, and its response to precipitation had a hysteresis effect. From the perspective of spatial variation, the influence of coal mining disturbance on surface soil moisture was not obvious. From the perspective of time series change, moving from wet to dry conditions, the soil in the disturbed area dried faster than the soil in the undisturbed area after soil wetted. When moving from drying to wetting, the soil in the disturbed area was quickly wetted. The EOF analysis showed that most observed spatial variability of soil moisture was stable in time. The study was conducted in a disturbed area and an undisturbed area for single EOF analysis, and the results showed that the EOF mode of the disturbed area was closer to that of the whole study area. By comparing the two subregions and the entire study area, it was found that the changes of correlation values were related to soil texture, bulk density, altitude and slope, indicating that the soil texture of the two subregions may be different at different elevations, and may also be related to the change of the original soil structure in the disturbed area. Overall, the EOF mode of the disturbed area determined the EOF mode of the entire study area.

foundation of Shanxi Province (201801D221103). The funder of the National Key Research and Development Program of China (2018YFD0200401) had no role in study design, data collection and analysis, decision to publish, or preparation of the manuscript. The funder of the Natural Science foundation of Shanxi Province (201801D221103) supported writing the paper.

**Competing interests:** The authors have declared that no competing interests exist.

# 1 Introduction

Soil moisture plays an important role in Earth's surface and atmospheric transport. It is an important variable in the global water and energy cycles. It is also an important research indicator for drought monitoring and crop yield estimation. It has a substantial impact on agriculture, hydrology, meteorology, and other fields. Traditional soil moisture monitoring methods mainly use the dry weight method and neutron moisture meter method for field measurements. These methods can only obtain soil moisture information at limited points, and the monitoring accuracy is affected by the sample density and spatial distribution, which is time-consuming, laborious, and costly. There is a need to achieve real-time and dynamic monitoring of soil moisture in a wide range of conditions and achieve certain accuracy requirements. The rapid development of remote sensing technology provides an effective means for accurate monitoring of surface soil moisture with high repetition coverage and regional scale. Microwave remote sensing has the advantage of not being limited by weather conditions and being sensitive to surface soil moisture changes, so it is widely used to monitoring soil moisture changes in arid and semi-arid areas [1–6].

As we know, the most widely used soil moisture inversion methods are a neural network algorithm (NN) [7–11], the water cloud model (WCM) [12–16], and a change detection algorithm (CD) [17–19]. When multi-temporal SAR data are available, the change detection (CD) method can be used in the absence of prior knowledge of the study area. Based on temporal SAR data, the relationship between soil moisture and the radar backscattering can be used to construct a coefficient by removing the influence of surface roughness. Based on the CD method, Zribi et al (2014). drew a soil moisture map of a semi-arid area by using ASAR satellite data [20]. The RMSE of the semi-arid area was 0.13 (soil moisture was about 0.035 m$^3$ / m$^3$). This method assumed that the change of vegetation and soil roughness had little effect on the change of the backscattering coefficient, and the change of the backscattering coefficient was mainly determined by the change in soil moisture. Based on Sentinel-1 radar data and Sentinel -2 optical data [21]. Reza A (2018) used the difference of backscattering coefficient to inverse the difference in soil water content. The inversion results were compared with the measured soil moisture, which verified the potential of the CD algorithm to inverse soil moisture under vegetation cover. Wickel et al. (2001) used temporal RADARSAT data to monitor soil water content in a wheat field after harvest, and established the relationship between soil water change and changes in radar backscattering [22].

In practice, there are few datasets of observational soil moisture because soil water content varies significantly with geographical location, time, and soil depth. Precipitation events and infiltration processes distribute soil moisture in a highly variable pattern throughout the basin. After one precipitation event, soil moisture is further redistributed through evaporation, transpiration, lateral flow and groundwater recharge. Usually, traditional artificial soil moisture measurements have high precision, but they are not practical for obtaining changing in soil moisture over time. To compensate for this deficiency, many scholars have developed many downscaling or interpolation methods to estimate soil moisture. These methods cover statistical methods and physical model-based inversion methods. Since it is relatively simple to access topographic, soil, and land use data can, understanding the dependence of soil moisture on these characteristics is particularly useful for interpolation and downscaling [23–26]. However, using reliable downscaling and interpolation methods requires a reasonable understanding of the variables controlling soil moisture patterns within the relevant spatial scales [27–30]. Ultimately, the complexity of soil moisture distribution patterns makes it difficult to estimate or predict them at unobserved times [31]. The EOF method is an effective tool for detecting spatiotemporal patterns of large multidimensional data sets This method has been widely applied

in the fields of meteorology, geology, and hydrology for analyzing large multidimensional data sets [32, 33]. Recently, there are also increasing interests in applying the EOF method to analyze spatial structures of soil moisture. Perry and Niemann [2007] identified two main spatial structures of soil moisture in Tarrawarra catchment, Australia, and showed that those spatial structures were mainly correlated with local soil properties and topographic attributes [34]. Jawson and Niemann [2007] found one primary spatial structure based on the soil moisture data of field activities in the southern Great Plains (SGP97) in 1997, which explained 61% of the total spatial variation of soil moisture and was highly correlated with soil texture [35]. Yoo and Kim [2004] analyzed the soil moisture data at two sites in SGP97 activity, and showed that terrain related factors mainly controlled the spatial structure of soil moisture [36]. Joshi and Mohanty [2010] analyzed the near surface soil moisture data at three different spatial scales, and found that the number of statistically significant spatial structures varied from four at the field scale to two at the watershed scale [37].

The Changhe River Basin is located in the Loess Plateau. The region is relatively arid with limited precipitation and water resources, coupled with the development of regional coal industry, consumes much of groundwater resources. Secondary geological disasters caused by coal mining subsidence, surface cracks and other issues, are exacerbated by the evaporation of regional water. High-intensity underground mining in the mining area will lead to surface subsidence and destroy the underground aquifer, which will directly lead to changes in the spatial distribution of soil moisture. Surface soil water directly affects the growth of surface vegetation. The terrain of the study area fluctuates greatly, and the layout of traditional soil moisture monitoring sampling points is difficult to reveal the large-scale change characteristics of the study area. Remote sensing monitoring technology can effectively and timely obtain regional soil water information.

At present, there are few studies on the temporal and spatial changes of soil surface moisture under the influence of mining subsidence. The soil moisture inversion method based on change detection algorithm does not need to measure the parameters such as surface roughness and vegetation. It only needs long-time radar data and optical data to effectively remove the influence of surface roughness and vegetation. In this study, SAR data were used to estimate the surface soil water over an extended time series in the study area using the CD algorithm based on radar signals over time. The temporal and spatial variation of surface soil water inside and outside the disturbed areas were compared and analyzed. To better understand potential spatial variation of soil water in the unobserved time, an empirical orthogonal function analysis method was used to analyze the potential temporal and spatial variation of surface soil water in the Changhe River Basin, and to determine the soil water regional distribution and variation in areas disturbed by coal mining. This will have reference significance for disaster control and ecological environment restoration in mining areas, and provide reliable decision support and scientific basis for the coordinated development of hilly areas of the Loess Plateau.

## 2. Materials and methods

### 2.1 Study area

The study area was located in Jincheng City in southeast Shanxi Province (112° 37′ 39″ - 112° 46′ 13″ E, 35° 30′ 14″ - 35° 38′ 04″ N). The site is 17.35 km wide from east to west, 22.47 km long from north to south, with a total area of 389.85 km$^2$ (Fig 1). The area contains mountains, with a mean altitude of 847 m a.s.l., ranging from 644–1193 m a.s.l. The mountainous areas are mainly distributed along the west and southeast of the Changhe River Basin. The distribution of hills is more extensive than that of mountains, and they account for more than half of the

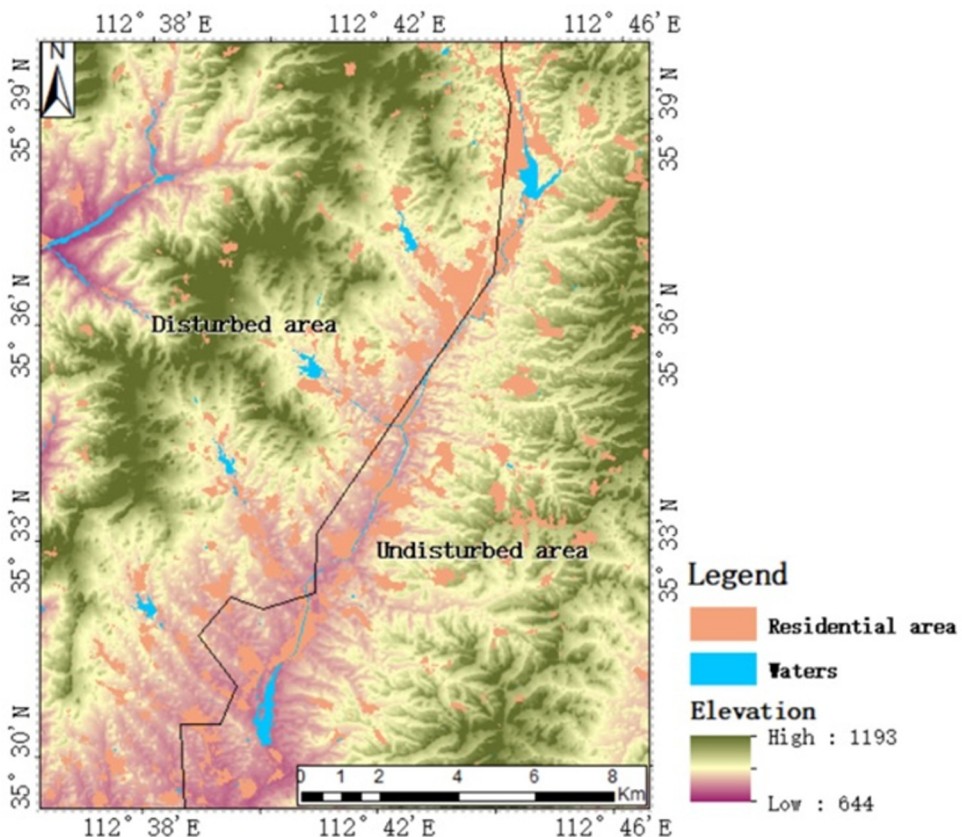

**Fig 1. Digital elevation model of study area.**

area of the entire region. The hills are mainly distributed on both sides of the long river and have developed into a dendritic shape, which are typical of the natural geographical characteristics of small watersheds in the loess hilly area. The region belongs to the mid latitude region, and it has a continental monsoon climate, which is warm and semi-humid in summer, cold and dry in winter, with precipitation concentrated in summer, but with less rain during spring and autumn [38]. The study area is divided into East and west areas by Changhe river. There are coal resources underground in the western region. Large area of surface damage is caused by coal mining. People have implemented large-scale land remediation on the damaged land, so we defined this region as a disturbed area. There is no coal resource underground in the eastern region, which we called the undisturbed area.

## 2.2 Radar signal change detection algorithm

A change detection algorithm is a method to estimate soil moisture using active microwave data for change detection based on single-band and multi-temporal radar images, which was first proposed by Wagner [39]. This method assumes that the scale of temporal variation of surface roughness and vegetation biomass is larger than that of soil moisture. Therefore, in the case of multi-temporal data, the change of radar backscattering intensity is caused by the change of soil moisture. In this way, in the multi-temporal radar data set, the influence of surface roughness and vegetation biomass is minimized, and the sensitivity of backscattering coefficients to represent changes in soil moisture is greatly improved. When using a long time series and, assuming adjacent transit times for two radar signals, the changes in farmland

roughness are small, and vegetation and soil moisture changes are directly related to the change in the backscattering coefficient.

Surface scattered signals received by radar include bare soil scattering radar signals attenuated by vegetation effects and vegetation scattering signals, which can be presented as follows:

$$\sigma_{sum}^0 = \sigma_{veg}^0 + \gamma^2(\theta)\sigma_{soil}^0 \tag{1}$$

$$\gamma^2(\theta) = \exp\left[-2\tau/\cos(\theta)\right] \tag{2}$$

where $\sigma_{sum}^0$ is the total backscattering coefficient received by radar, $\sigma_{veg}^0$ is the backscattering coefficient that is contributed by vegetation, $\gamma^2(\theta)$ is the attenuation coefficient of the vegetation canopy, $\theta$ is the incident angle of radar, $\tau$ is the optical thickness related to the geometric structure and water content of the vegetation.

Under a given NDVI condition, the difference between the backscattering coefficient and the minimum backscattering coefficient of the cell (i, j) at date d in the time series can be presented as follows:

$$\Delta\sigma_{(i,j)}^{NDVI} = \sigma_{(i,j),ndvi}^0(d) - \sigma_{dry(i,j),ndvi}^0 = f_{(i,j)}(NDVI, Mv) \tag{3}$$

where $\sigma_{(i,j),ndvi}^0(d)$ is the backscattering coefficient of a pixel under a given NDVI condition, $\sigma_{dry(i,j),ndvi}^0$ is the minimum backscattering coefficient of the cell under the given NDVI condition in the time series, and $f_{(i,j)}(NDVI,Mv)$ is a function related to soil moisture content and NDVI.

The studies of Baghdadi (2007) and Srivastava et al (2009). showed a linear relationship between the difference of radar signals and the change of soil moisture under bare soil or vegetation conditions [40, 41], and the relationship can be presented as follows:

$$\frac{Mv - Mv_{min}}{Mv_{max} - Mv_{min}} = \frac{\sigma^0 - \sigma_{min}^0}{\sigma_{max}^0 - \sigma_{min}^0} \tag{4}$$

where Mv is the soil water content in a time series, $Mv_{max}$ is the maximum soil water content in a time series. $Mv_{min}$ is the minimum soil water content in a time series. $\sigma^0$ is the backscattering coefficient of a certain time in a time series, $\sigma_{max}^0$ is the maximum backscattering coefficient in a time series, and $\sigma_{min}^0$ is the minimum backscattering coefficient in time series.

There is a linear relationship between the difference in radar signals and the change in soil moisture. Under a given NDVI condition, the change of backscattering coefficient is determined by the change in soil moisture:

$$\Delta\sigma_{NDVI}^0 = a(NDVI)\Delta Mv \tag{5}$$

where $\Delta Mv$ is the variation of soil moisture in a pixel at date d and the minimum soil moisture in the time series under the given NDVI condition. $\Delta\sigma_{NDVI}^0$ is the variation of the backscattering coefficient of a pixel at date d and the minimum backscattering coefficient in the time series under the same NDVI condition. Parameter a depends on NDVI, representing the impact of vegetation. When NDVI increase, the sensitivity of radar signal to soil water content gradually weakens, and the change of the backscattering coefficient decreased with the increase of NDVI. $\Delta\sigma_{NDVI}^0$ was negatively correlated with NDVI (Fig 2).

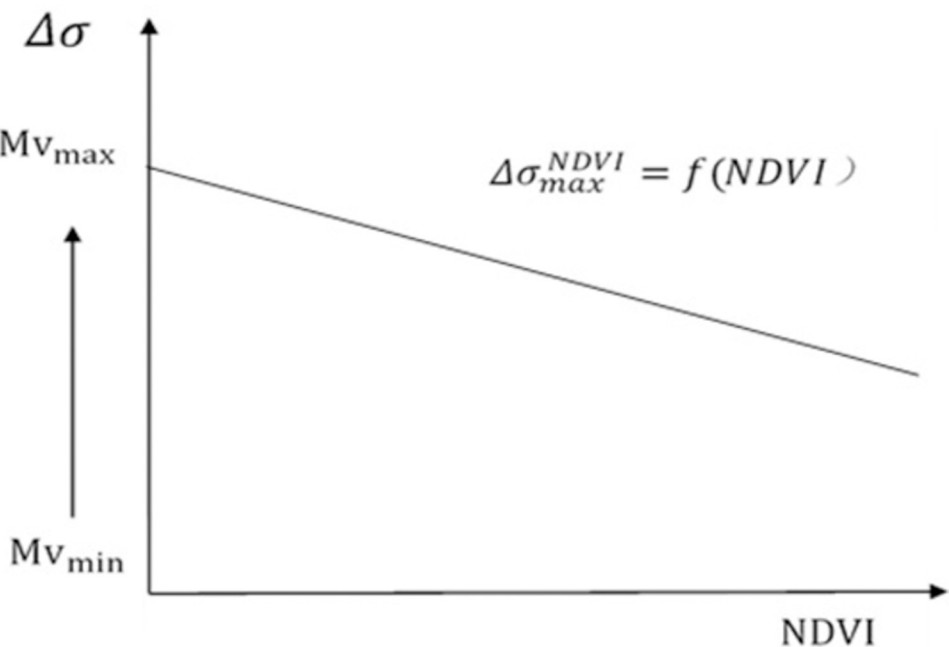

**Fig 2. Relationship between backscattering coefficient variation and NDVI.**

According to Eq (5), under certain NDVI conditions, the maximum variation of the backscattering coefficient is determined by the maximum variation of soil water content:

$$\Delta\sigma_{max}^{NDVI} = a(NDVI)\Delta Mv_{max} \qquad (6)$$

$\Delta Mv_{max} = Mv_{max} - Mv_{min}$. $Mv_{max}$ and $Mv_{min}$ are constants, so $\Delta\sigma_{max}^{NDVI}$ can be expressed by NDVI:

$$\Delta\sigma_{max}^{NDVI} = f(NDVI) = a * NDVI + \Delta\sigma_{max}^{baresoil} \qquad (7)$$

Under the condition of a given NDVI:

$$\Delta\sigma_{max}^{NDVI} = \sigma_{max}^{NDVI} - \sigma_{min}^{NDVI} \qquad (8)$$

$\Delta\sigma_{max}^{baresoil}$ is the maximum change in the backscattering coefficient when NDVI is 0, that is, in bare soil.

According to the equations, the soil moisture of each cell can be obtained by the following equation:

$$M_{Vd}^{(i,j)} = \frac{\sigma_d^{(i,j)NDVI} - \sigma_{min}^{(i,j)NDVI}}{f(NDVI)}(Mv_{max} - Mv_{min}) + Mv_{min}^{(i,j)} \qquad (9)$$

where $M_{Vd}^{(i,j)}$ is the soil moisture content of the cell, $\sigma_d^{(i,j)NDVI}$ is the backscattering coefficient of the cell at time d, and $\sigma_{min}^{(i,j)NDVI}$ is the minimum backscattering coefficient of the cell in the time series.

## 2.3 Data processing

**(1) Sentinel-1 GRD data.** The Sentinel-1 satellite database corresponds to the period from January 2018 to December 2019 (Table 1). The pre-processing of 24 Sentinel-1 GRD

**Table 1. Parameters of Sentinel-1 GRD data.**

| Imaging time | Incident angle |
|---|---|
| 20180125, 20180218, 20180314, 20180419, 20180513, 20180618 | 39˚ |
| 20180712, 20180817, 20180910, 20181028, 20181121, 20181215 | |
| 20190120, 20190213, 20190321, 20190414, 20190520, 20190601, 20190707, 20190812, 20190917, 20191011, 20191116, 20191210 | |

images was completed Using SNAP software. According to previous studies, by comparing the inversion results of VV and VH data, the backscattering coefficient of VV polarization was more sensitive to the difference in soil water content, and the soil water accuracy inverted by VV polarization data was significantly higher than that inverted by VH. Therefore, the backscattering coefficients of the study area were extracted by VV polarization. After noise removal, calibration and terrain correction, the Lee filtering algorithm with window size of 5X5 was used for speckle filtering. Finally, the backscattering coefficient image with geographical coordinate information was generated with geographical coding. Although the spatial resolution of Sentinel-1 data is 10 m, in order to reduce the uncertainty caused by soil roughness and uneven vegetation coverage, the data were resampled to the cell size of 50 m.

**(2) Sentinel-2 data.** The Sentinel-2 data obtained was consistent with or similar to Sentinel-1 data without cloud cover (Table 2). The Sentinel-2 data corresponds to images recorded in 13 spectral bands, with a spatial resolution of 10 m. In the present study, band 4 (Red) and band 8 (NIR) are used to calculate the NDVI.

**(3) Ground measurements.** The acquisition time of data used in this study was October 28,2018 and August 12, 2019, which is consistent with the imaging time of the Sentinel-1 satellite. The sampling points were selected from 12 farmlands in different locations. Time-domain reflectometry was used to collect the average value of five times at each sampling point, and GPS positioning was used to record the coordinates of the samples.

**(4) Soil texture and topographic attributes.** The digital elevation model (DEM) of the study area adopted ASTER GDEM. The factors such as slope, curvature, terrain roughness, natural logarithm of catchment area (Ln Area) and terrain wetness index (WTI), were extracted from the DEM of the study area. Soil texture such as silt, sand, clay and bulk density (BD) and precipitation data were obtained from the National Earth System Science Data Center (China).

**(5) Estimation of soil moisture.** By counting the backscattering coefficient of each cell, the minimum value of the backscattering coefficient of each cell in the time series was obtained, and then the scattering coefficient of each period was subtracted from the minimum value to obtain the scattering coefficient difference for the time series. Finally, the statistical analysis was performed with NDVI of the time series (Fig 3).

With a gradual increase of NDVI, the backscattering coefficient gradually decreased, and the f(NDVI) relationship was obtained by the upper decile regression statistical analysis [42]. Due to the lack of long-term ground measurement of soil water content in the study area, this

**Table 2. Sentinel-2 data.**

| Imaging time |
|---|
| 20180101, 20180212, 20180314, 20180428, 20180521, 20180622, 20180928, 20181028, 20181122, 20181217, 20190121, 20190319, 20190416, 20190521, 20190602, 20190707, 20190816, 20190918, 20191008, 20191119, 20191219 |

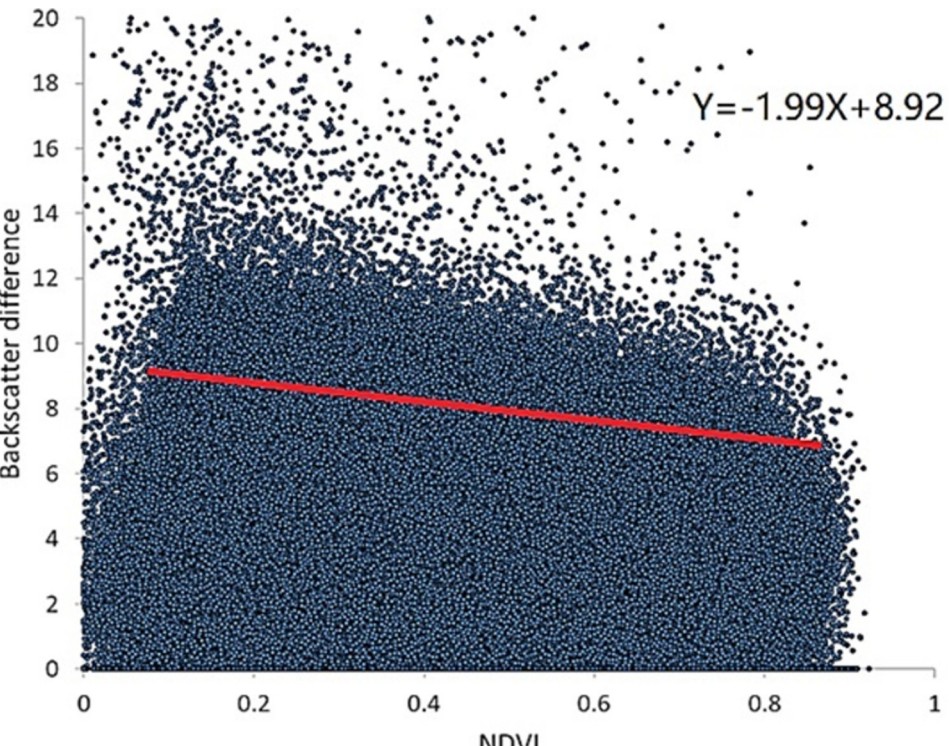

**Fig 3. Relationship between backscattering coefficient difference and NDVI.** Each point corresponds to the radar signal difference and NDVI in the same pixel position.

paper used SMOS products corresponding to Sentinel-1 data time to estimate the maximum and minimum soil water content in the time series of the study area, which were $Mv_{max}$ = 0.29cm³ / cm³ and $Mv_{min}$ = 0.01cm³ / cm³, respectively. The backscattering data of 24 dates and the corresponding NDVI data were inputted into Eq (9), and 24 soil moisture distribution maps for the study area were obtained.

**(6) EOF calculation.** EOF analysis is a statistical method widely used to analyze large multi-dimensional data sets. Normally, the EOF analysis method can be divided into spatial and temporal modes. Each mode is composed of a spatial pattern (the so-called EOF) and a principal component (PC) time series that represents the temporal evolution of the EOF pattern [43, 44]. A given mode can be reconstructed by multiplying the EOF (space) by its PC (time). The significance of the modes was evaluated by computing their sampling error as $\lambda$(2/N)1/2, where $\lambda$ is a given eigenvalue and N the number of realizations. A detailed description about EOF analysis of soil moisture can be found elsewhere [45–47] and only a brief discussion is offered here.

For a soil moisture data set with n locations and m observations at each location, spatial anomalies of soil moisture can be computed as

$$Z_i(t) = S_i(t) - \frac{1}{m}\sum_{j=1}^{m}S_j(t) \qquad (10)$$

where $Z_i(t)$ and $S_i(t)$ are soil moisture spatial anomaly and volumetric soil moisture content at location i and time t, respectively. A matrix of soil moisture spatial anomalies, Z, can be

constructed as

$$Z = \begin{bmatrix} Z_{11} & \cdots & Z_{1n} \\ \vdots & \ddots & \vdots \\ Z_{m1} & \cdots & Z_{mn} \end{bmatrix} \tag{11}$$

Then, a covariance matrix V can be calculated as

$$V = \frac{1}{m-1} Z^T Z \tag{12}$$

where the superscript T indicates matrix transpose. To perform EOF analysis, one needs to find eigenvectors and eigenvalues for V, which satisfy the following equation:

$$VE = LE \tag{13}$$

where E contains eigenvectors in columns

$$E = \begin{bmatrix} e_{11} & \cdots & e_{1n} \\ \vdots & \ddots & \vdots \\ e_{n1} & \cdots & e_{nn} \end{bmatrix} \tag{14}$$

and L contains eigenvalues along the diagonal

$$L = \begin{bmatrix} l_{11} & \cdots & 0 \\ \vdots & \ddots & \vdots \\ 0 & \cdots & l_{nn} \end{bmatrix} \tag{15}$$

The above procedure rotates the original coordinate axes with each axis indicating a sampling time into a new set of orthogonal coordinate axes with each eigenvector representing a new axis. The eigenvalues explain the variance in the data along the direction of each corresponding new axis, and the portion of the explained variance ($P_j$) by the jth new axis in the total variance can be computed as

$$P_j = \frac{l_{jj}}{\sum_{k=1}^{n} l_{jj}} \tag{16}$$

The eigenvectors are then arranged according to eigenvalues: the first axis explains the largest variance in the data, while each following axis explains the largest remaining variance and is orthogonal to other axes. F can be computed by projecting Z onto E

$$F = ZE \tag{17}$$

The purpose of EOF analysis is to reduce the dimensionality of a data set, and the approach of North et al. adopted to select statistically significant EOFs. This approach assumes that an EOF is statistically significant if the lower confidence limit (e.g., 95%) of its eigenvalue is greater than the upper confidence limit of the remaining largest eigenvalue. The 95% confidence interval for the kth eigenvalue can be calculated as

$$CL_k = \lambda_k \times \left( 1 \pm \sqrt{\frac{2}{m}} \right) \tag{18}$$

where $CI_k$ is the confidence interval for the kth eigenvalue and m is the number of sampling locations.

## 3. Results and discussion

### 3.1 Validation of soil moisture retrieved by SAR

The 12 measured soil moisture data points at a depth of 5 cm were used to verify the accuracy of the soil moisture data retrieved by SAR at the corresponding date. Fig 4 is the scatter plot of the comparison between the measured soil moisture and the retrieved value. The $R^2$ of the two dates are 0.7246 and 0.6799, and the RMSE are 0.052 and 0.049, respectively. The results showed that the soil water content estimated by the time series radar signal CD algorithm in the study area was accurate.

### 3.2 Traditional analysis of soil moisture

The statistical characteristics of soil moisture at 24 dates is shown in Table 3. The minimum (0.06) and maximum (0.21) of the mean value appeared on February 13,2019, and September 17, 2019, respectively. The minimum (0.32) and maximum (0.69) coefficients of variation appeared on June 6, 2018 and February 13, 2019, respectively. The maximum skewness and kurtosis (1.52 and 3.97) appeared on February 13, 2019.

The study area is located in the semi-arid region of the Loess Plateau, which has deep groundwater levels. The soil moisture in the four seasons of the year shows a dry-wet-dry variation (Fig 5). Fig 6(A) is the comparison chart of the average soil moisture and daily precipitation in two years. The soil moisture varied seasonally. The average soil moisture in January–March 2018, October–December 2018, and January–March 2019, November–December 2019 were below 15%, and the daily precipitation at the corresponding time was below 10 mm. In 2018, the precipitation increased significantly from April to September, and the daily precipitation was greater than 20 mm during this period. The corresponding soil moisture increased from April, and decreased with a decrease in precipitation after reaching the peak in June. In 2019, the precipitation significantly increased from April to October, and fluctuated from May to July. It reached the peak of 54 mm in August 3, and gradually decreased from September to October. The corresponding soil moisture started increasing in April, and gradually decreased after reaching the peak in September. This indicated that the surface soil moisture in the study area was significantly affected by precipitation factors, and the response in soil moisture lagged precipitation.

Fig 6(B) is a comparison of coefficient of variation of soil moisture and daily precipitation in two years. The coefficient of variation in soil moisture varied by seasons. The changes in

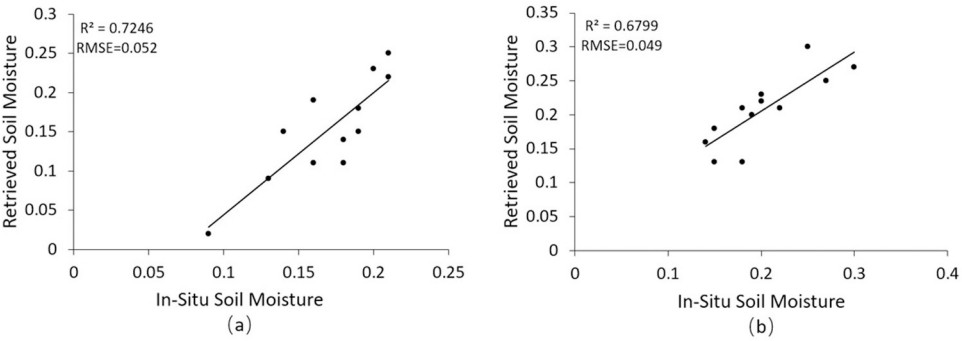

**Fig 4. Comparison of measured and estimated soil moisture.** (a) October 28,2018, and (b) August 12, 2019.

**Table 3. Classical statistics of soil moisture.**

| Time | Min. | Max. | Avg. | Median | Std. | Skewness | Kurtosis | Variation |
|---|---|---|---|---|---|---|---|---|
| 20180125 | 0.01 | 0.47 | 0.08 | 0.07 | 0.05 | 1.00 | 1.26 | 0.65 |
| 20180218 | 0.01 | 0.43 | 0.12 | 0.11 | 0.05 | 0.63 | 0.87 | 0.43 |
| 20180314 | 0.02 | 0.31 | 0.10 | 0.10 | 0.04 | 0.37 | 0.27 | 0.44 |
| 20180419 | 0.02 | 0.45 | 0.11 | 0.11 | 0.05 | 0.32 | 0.47 | 0.45 |
| 20180513 | 0.02 | 0.47 | 0.13 | 0.13 | 0.05 | 0.30 | 0.59 | 0.38 |
| 20180606 | 0.03 | 0.48 | 0.20 | 0.20 | 0.07 | 0.13 | 0.02 | 0.32 |
| 20180712 | 0.02 | 0.50 | 0.19 | 0.19 | 0.07 | 0.21 | -0.04 | 0.35 |
| 20180817 | 0.01 | 0.41 | 0.16 | 0.15 | 0.06 | 0.35 | 0.12 | 0.37 |
| 20180922 | 0.02 | 0.46 | 0.14 | 0.13 | 0.06 | 0.43 | 0.34 | 0.41 |
| 20181016 | 0.02 | 0.41 | 0.12 | 0.12 | 0.05 | 0.48 | 0.62 | 0.40 |
| 20181121 | 0.02 | 0.45 | 0.12 | 0.11 | 0.05 | 0.58 | 1.20 | 0.42 |
| 20181215 | 0.01 | 0.43 | 0.09 | 0.08 | 0.04 | 0.82 | 2.17 | 0.48 |
| 20190120 | 0.01 | 0.36 | 0.07 | 0.07 | 0.04 | 0.91 | 1.61 | 0.56 |
| 20190213 | 0.01 | 0.37 | 0.06 | 0.05 | 0.04 | 1.52 | 3.97 | 0.69 |
| 20190321 | 0.01 | 0.47 | 0.08 | 0.08 | 0.04 | 0.74 | 1.54 | 0.53 |
| 20190426 | 0.02 | 0.43 | 0.11 | 0.11 | 0.05 | 0.58 | 1.52 | 0.46 |
| 20190520 | 0.02 | 0.42 | 0.12 | 0.12 | 0.05 | 0.33 | 0.41 | 0.40 |
| 20190625 | 0.02 | 0.45 | 0.13 | 0.13 | 0.05 | 0.44 | 1.00 | 0.38 |
| 20190719 | 0.02 | 0.49 | 0.13 | 0.12 | 0.05 | 0.61 | 1.19 | 0.41 |
| 20190812 | 0.02 | 0.49 | 0.18 | 0.18 | 0.06 | 0.32 | 0.22 | 0.35 |
| 20190917 | 0.03 | 0.49 | 0.21 | 0.21 | 0.08 | 0.18 | -0.47 | 0.37 |
| 20191023 | 0.02 | 0.49 | 0.20 | 0.19 | 0.08 | 0.23 | -0.34 | 0.40 |
| 20191116 | 0.02 | 0.46 | 0.13 | 0.12 | 0.05 | 0.66 | 1.30 | 0.42 |
| 20191222 | 0.02 | 0.39 | 0.10 | 0.10 | 0.05 | 0.69 | 1.06 | 0.48 |

variation coefficients showed trends that were the opposite of the temporal characteristics of soil moisture. For example, the variation coefficient was small on wet days, such as June 6, 2018 and August 12, 2019, and the variation coefficient on a dry date was high, such as January 25, 2018, and February 13, 2019. This indicated that in the period of drought, due to less precipitation, soil moisture was mainly affected by vegetation. The cultivated land in the study area was mainly planted with winter wheat, and the vegetation of forests and grasslands in this period was lower. These differences in vegetation coverage caused differences in soil moisture during the drought period. In the wet period, due to the strong influence of precipitation, the difference in soil moisture were small.

### 3.3 The characteristics of soil moisture in the disturbed coal minging areas

Based on the spatial extent of the coal mining activities, the study site was divided into a disturbed area and an undisturbed area. The spatial average soil moisture in the two areas were statistically analyzed (Fig 7). The results showed that the soil moisture in the disturbed area was lower than that in the undisturbed area in 16 days of 24 days, accounting for 67% of the total data, which indicated that coal mining disturbance had a negative impact on surface soil moisture, but the effect of subtle. It is worth noting that, from the perspective of temporal changes, the soil water content in the disturbed area was slightly lower than that in the undisturbed area when soil conditions were changing from wet to dry. For example, between June 2018 and February 2019 (the area between the two broken lines in this period is 0.6), and between September 2019 and December 2019 (the area between the two broken lines in this

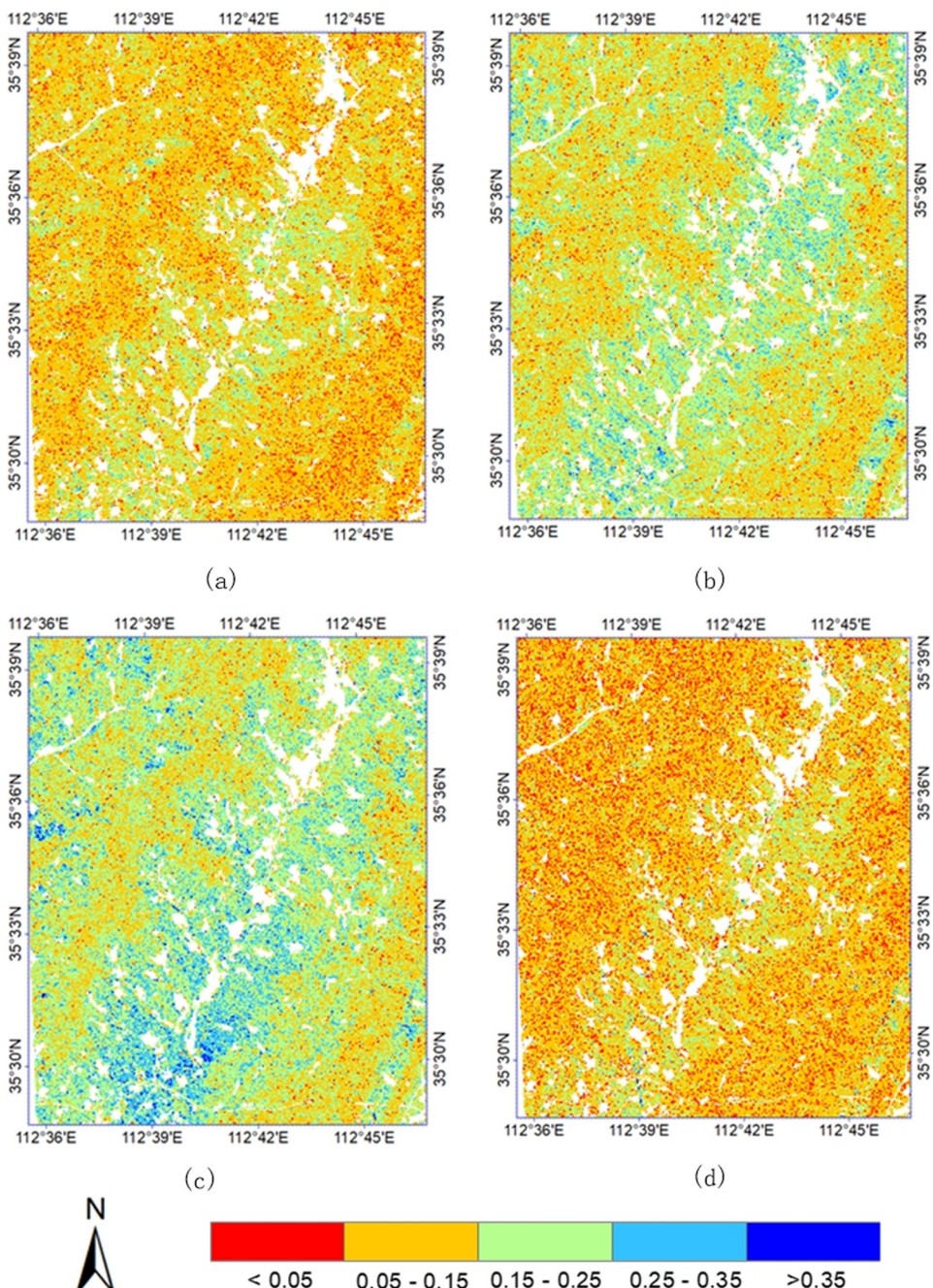

**Fig 5. Soil moisture map estimated based on SAR data.** (a) February 18,2018, (b) August 17, 2018, (c) August 12, 2019, and (d) December 10, 2019.

period is 0.27) indicates that the soil in the disturbed area had dried faster than the soil in the undisturbed area. In the process of moving from dry to wet condition, the soil moisture in the disturbed area was slightly higher than in the undisturbed area. This can be observed between March 2018 and June 2019 (the area between the two broken lines during this period is 0.252), and between March 2019 and May 2019 (the area between the two broken lines during this period is 0.186), and this indicates that the soil moisture in the disturbed area quickly

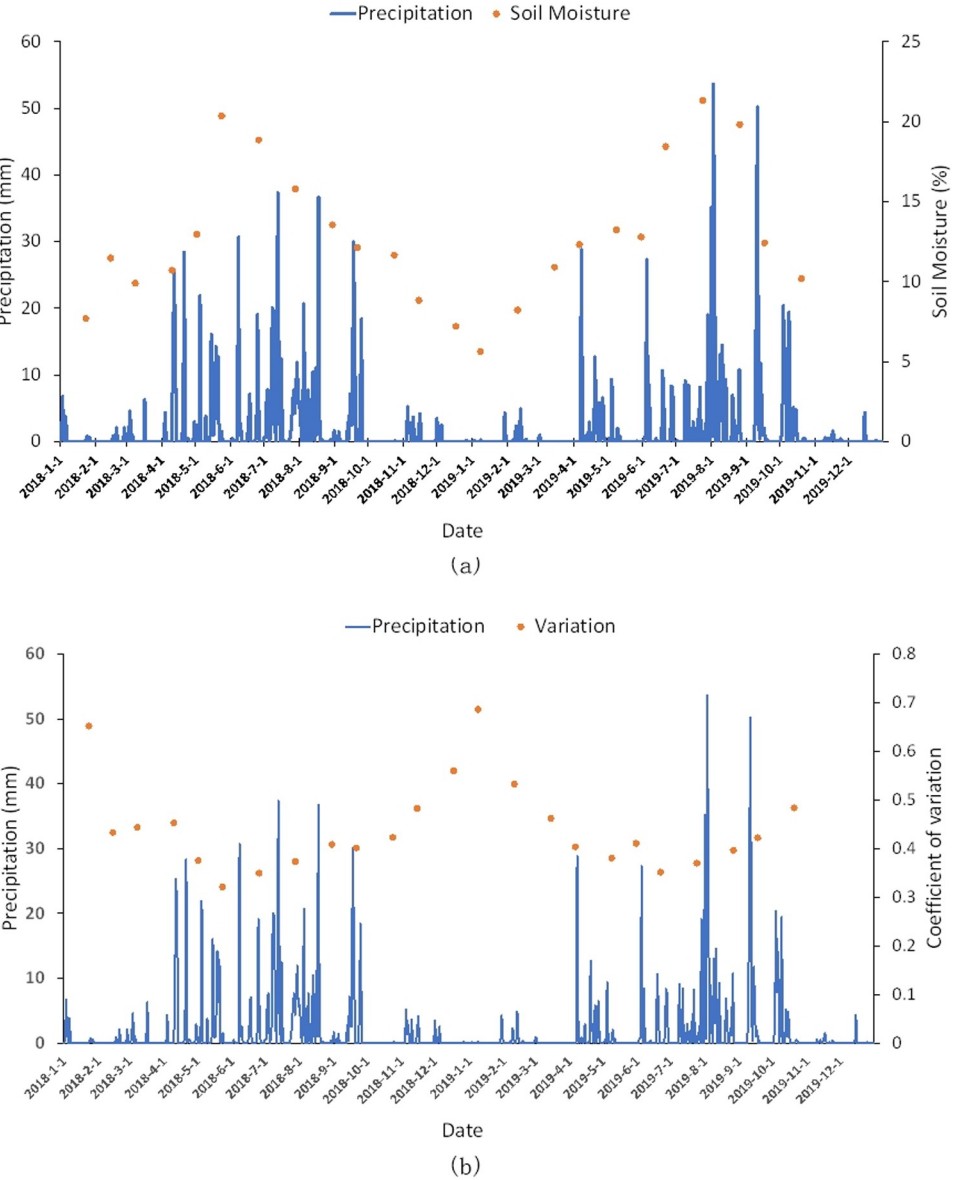

**Fig 6. Comparison between soil moisture and monthly precipitation.** (a) average of soil moisture, (b) coefficient of variation of soil moisture.

recharged in this process. The characteristics of this temporal change may be due to surface cracks, collapses, and excavations caused by coal mining in disturbed area, and reclamation measures such as covering soil in some areas, thereby increasing soil porosity and changing soil moisture holding capacity.

## 3.4 Spatial and temporal characteristics of soil moisture based on EOF

Twenty-four modes were obtained by EOF analysis of soil moisture in the study area. Table 4 shows the variance contribution rate of the first six EOFs, and the cumulative variance contribution rate is 75.9%. The upper and lower confidence intervals of EOF1 and EOF2 are 75.8% and 41.8%, 9.1% and 5.0%, respectively. For EOF3, EOF4, and EOF5, the upper and lower

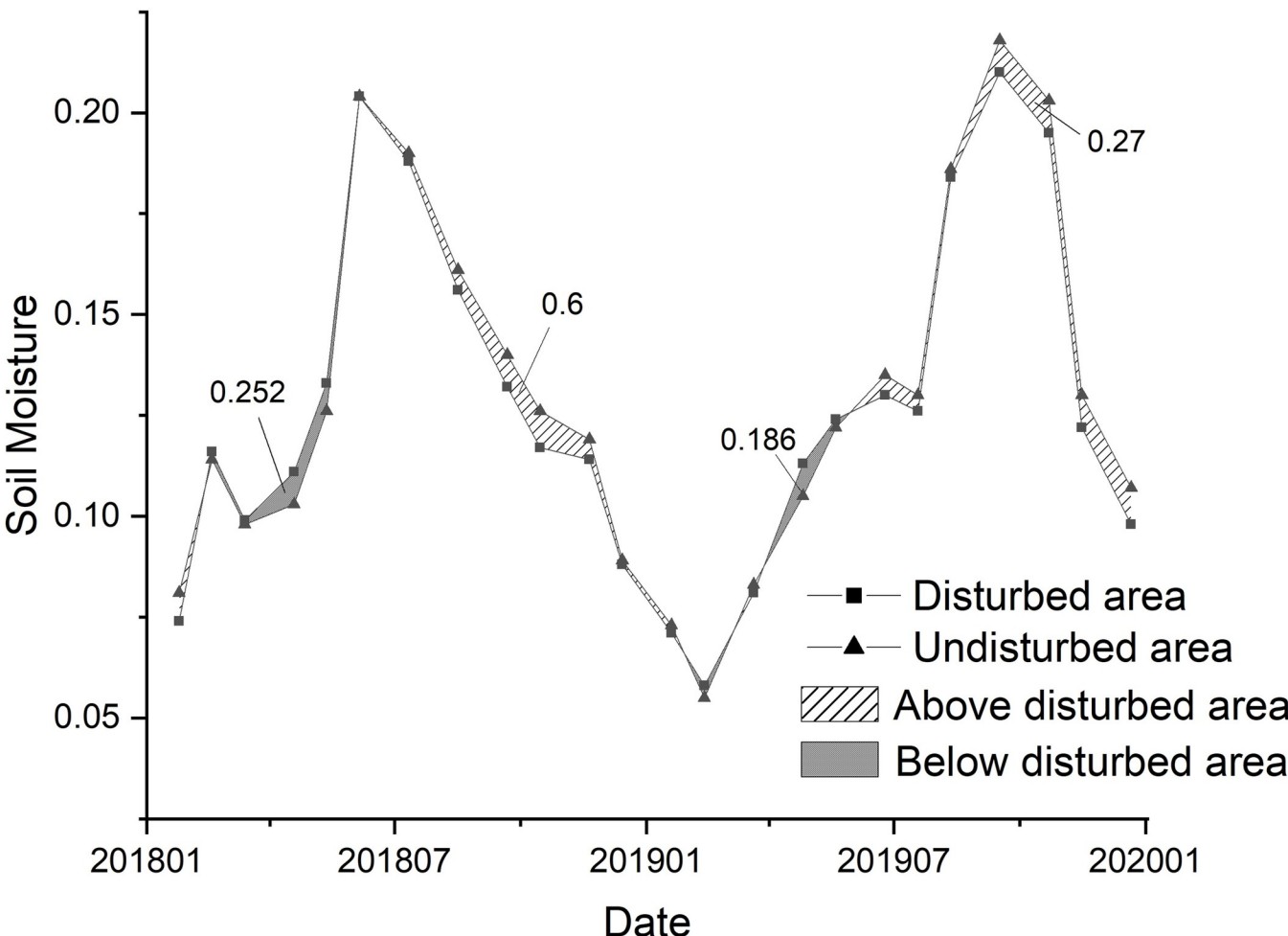

**Fig 7. Comparison of soil moisture time series between disturbed and undisturbed areas.**

confidence intervals are 3.9% and 2.2%, 3.2% and 1.8%, 3.0% and 1.6%, respectively. Only EOF1 which had a variance contribution rate of 58.8% and EOF2 with variance contribution rate of 7.1% were statistically significant. The two spatial patterns explained 65.9% variance of the data set, indicating that less potential spatial structure could explain the complex soil moisture distribution pattern in the study area. It is worth noting that although the remaining 22 spatial modes were not significant, the explanatory power is 34.1%, indicating that about one third of the spatial variability of soil moisture was random in time, that is, the spatial

**Table 4. Variance contributions (%) of the first six EOFs.**

| Mode | Explained variance | Cumulative explanatory variance | Eigenvalue confidence interval | |
|---|---|---|---|---|
| | | | Upper limit | Lower limit |
| 1 | 58.8% | 58.8% | 75.8% | 41.8% |
| 2 | 7.1% | 65.9% | 9.1% | 5.0% |
| 3 | 3.1% | 69.0% | 3.9% | 2.2% |
| 4 | 2.5% | 71.5% | 3.2% | 1.8% |
| 5 | 2.3% | 73.8% | 3.0% | 1.6% |

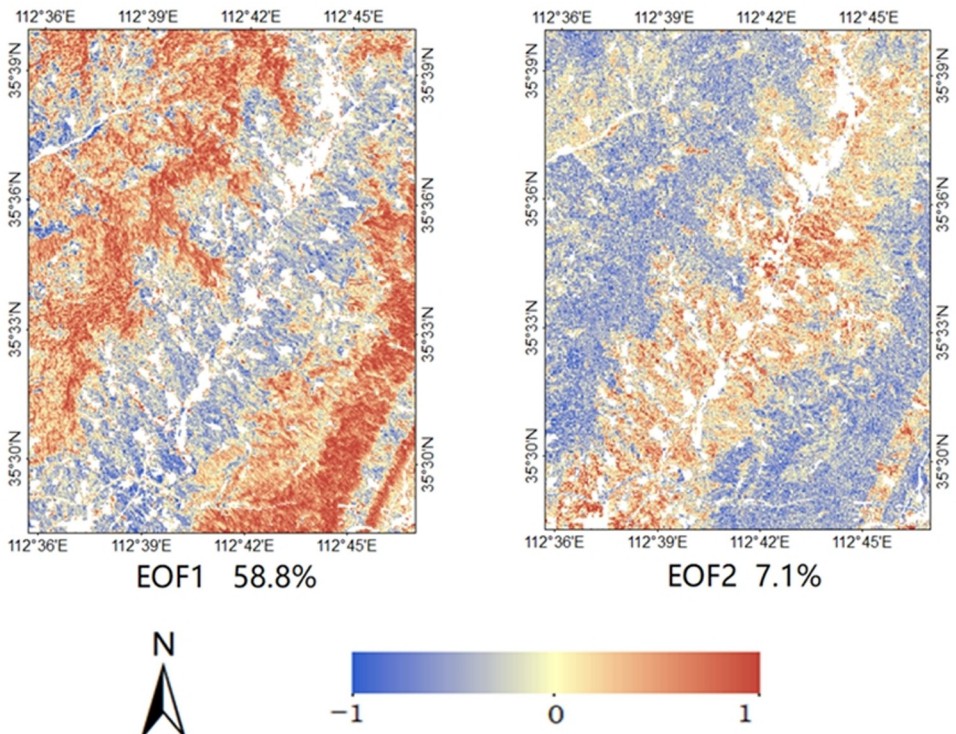

**Fig 8. The first two EOFs generated from the soil moisture and their explained variance.**

distribution is independent of time. Fig 8 shows the first two EOFs modes. Fig 9 shows the PC value and spatial average of soil moisture at each time.

Fig 8 shows that the spatial distribution of EOF1 is very similar to the topography of the study area, that is, the EOF value along both sides of the river presents a negative value, and the EOF value on the hillside presents a trend from 0 to positive. The spatial pattern of EOF1 is

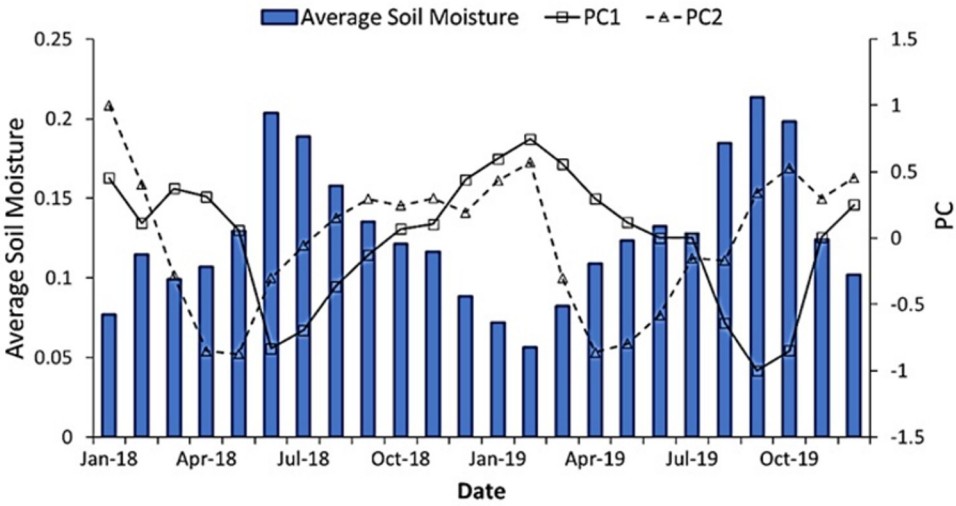

**Fig 9. The time series of the PC and spatial average soil moisture.**

similar to the soil moisture distribution pattern on June 6, 2018, September 17, 2019, and October 23, 2019. EOF1 mode had both positive and negative values, indicating inconsistent changes in soil moisture over time [48]. The average value of each cell of the soil moisture set in the time series was calculated to obtain the data Mw. The correlation analysis between EOF1 and data Mw showed that EOF was negatively correlated with Mw (−0.61). The EOF1 spatial mode was related to the dry–wet period (Fig 9). Usually, PC1 value was positive under dry conditions, but negative under wet conditions. When the regional mean soil moisture was low, it corresponded to a large PC1 value. Therefore, EOF1 showed low temporal variation in areas with low soil water. For example, when it was dry February 13, 2019, the PC1 value was 0.75. Considering the negative correlation between EOF1 and Mw, the product of EOF1 and PC1 was more strongly positive in the dry area, resulting in lower losses of soil moisture in the dry area. On June 6, 2018, the PC1 value (-0.83) resulted in a more negative product values of EOF1 and PC1, resulting in less increase in soil water at dry locations. Western et al. suggested that lateral redistribution was the main hydrological flux under moist conditions [49]. Water flows mainly through the shallow soil layer and relatively high permeability A soil layer, and the hydraulic gradient of these parts is controlled by topography. Although there were differences in the relative importance of EOF1 to the monthly variation of soil moisture during the dry-wet cycle, the spatial pattern of monthly variation of soil moisture were stable in time.

The main advantage of EOF analysis is that it uses less orthogonal spatial mode to explain most spatial variations of the soil moisture data. To determine the driving factors affecting the spatial differentiation of soil moisture, correlations between EOF and potential driving factors (topographic factors and soil properties) was analyzed (Table 5). The correlation between the two main spatial patterns and the potential factors were low. For example, the correlation with the natural logarithm and curvature of the catchment area was close to 0. The reason for this result may be related to the estimation of soil water in the study area by SAR data. Some regional characteristics may have also contributed to this, but they have limited impacts on irregular spatial patterns. The correlation between the two spatial modes and altitude, slope, and terrain roughness were high, and the correlation with sand, clay and bulk density were moderate. The results showed that the driving factors of variation in EOF1 spatial patterns

**Table 5. Correlations between EOFs and regional characteristics in sub-regions.**

|  | Whole area | | Disturbed area | | Undisturbed area | |
|---|---|---|---|---|---|---|
|  | EOF1 | EOF2 | EOF1 | EOF2 | EOF1 | EOF2 |
| Silt | 0.03* | 0.02* | 0.17* | 0.05* | -0.14** | -0.08* |
| Sand | -0.13** | 0.06** | -0.23** | -0.03** | 0.24** | 0.15** |
| Clay | 0.17** | -0.14** | -0.04** | -0.08** | 0.13** | -0.09** |
| Ln Area | 0.03** | -0.02* | 0.04** | -0.04* | 0.03** | -0.01* |
| Elevation | 0.42** | -0.35** | 0.60** | -0.23** | -0.01** | -0.16** |
| Curvature | -0.01 | -0.01 | -0.01 | -0.01 | 0.01 | -0.01 |
| Slope | 0.43** | -0.39** | 0.30** | -0.29** | 0.57** | -0.30** |
| Roughness | 0.39** | -0.32** | 0.26** | -0.23** | 0.29** | -0.14** |
| WTI | -0.15** | 0.16** | -0.08** | 0.08** | -0.18** | 0.10** |
| BD | -0.16** | 0.04** | -0.24** | -0.03** | 0.11** | 0.09** |
| Mw | -0.61** | 0.29** | - | - | - | - |

Note

*, significant correlation at P<0.05 level

**, significant correlation at P<0.01 level; Ln Area, Natural logarithm of catchment area

were related to soil properties and topographic characteristics, because when the soil bulk density, and the percentage of sand and clay were combined with altitude, slope, and other factors, the speed of soil water redistribution after rainfall was determined. Sandy soil drain quickly and were therefore drier than other regions. On the contrary, the clay area had a greater capacity for water retention and had higher humidity. The dominant effect of soil texture on soil moisture has been widely recognized by previous studies [50]. Tiejun Wang (2017) found that the primary spatial structure of soil moisture was mostly correlated with soil texture in all the study regions, indicating the dominant role of soil in determining soil moisture spatial variability [51].

In contrast, although EOF2 was characterized by basin topography as EOF1, EOF2 values were positive on both sides of the river and negative on the hillside. Most of the EOF2 values at lower altitudes in the northwest were also negative. In Fig 9, the PC2 value changed when there was a change in status in the dry-wet cycle, and the PC2 value also increased when moving from wet to dry. Different from the PC1 value, the PC2 value showed the minimum negative value in the medium humid period. Analysis showed that EOF2 was positively correlated with Mw (0.29), and negatively correlated with altitude and slope (-0.35 and -0.39, respectively). Taking the dates of February 13, 2019 (dry), and October 23, 2019 (wet), as examples, PC2 was positive (0.57 and 0.53, respectively). Therefore, the products of EOF2 and PC2 were larger at lower altitudes, which led to less soil moisture loss under dry conditions, leading to greater increases in soil moisture under wet conditions at lower altitudes. This may be due to the fact that wind speed in loess hilly areas is usually lower at lower altitudes. This maintains relatively high humidity and prevents excessive moisture loss due to evaporation or transpiration in lower areas [52]. EOF2 was negatively correlated with altitude and slope, which also shows that the soil water distribution is more dynamic at the location with higher altitude and steeper slope. This result is consistent with previous studies [53]. They found that the slope position is the largest contributor to the temporal variability of soil moisture. Steeper slopes are expected to promote drainage and produce higher temporal variability. Higher altitude areas may be more easily separated from stable water sources, such as rivers or regional aquifers, so the soil water change in this area is more dynamic.

According to the two modes, the EOF1 mode was dominant in the wettest date, and the EOF2 mode was dominant when soil water content was moderate. The two modes complemented each other and explain the potential trend of soil moisture.

The study area is divided into disturbed and undisturbed area to analyze the possible influence, and the EOF analysis of the two sub-regions was repeated by recalculating the spatial anomaly. If the spatial mean value varies significantly between two subregions, or if the temporal correlation used to identify EOF mainly occurs in one subregion or another, the new EOF mode may be very different. However, the EOF modes of the final two sub-regions were similar to those of the entire study area. The EOF1 mode in the disturbed area explained 58.1% of the variance, and the EOF2 mode explained 7.2% of the variance. The EOF1 mode in the undisturbed area explained 60.2% of the variance, and the EOF2 mode explained 6.8% of the variance. By contrast, the EOF mode of the disturbed area was closer to that of the whole study area. In addition, the PC values of the two sub-regions corresponding to the EOF mode were also similar to those of the whole study area.

The correlations between the EOF mode of the sub-region and the regional characteristics were analyzed (Table 5, Fig 10). The analysis showed that regardless of the mode, the correlation between the disturbed area and the whole area, and these three characteristics, was similar, but the undisturbed area had different results. For example, in EOF1 mode, the correlation between the disturbed area and percentage of sand particles was -0.23, and the whole area and the percentage of sand particles was −0.13. The correlation between the undisturbed area and

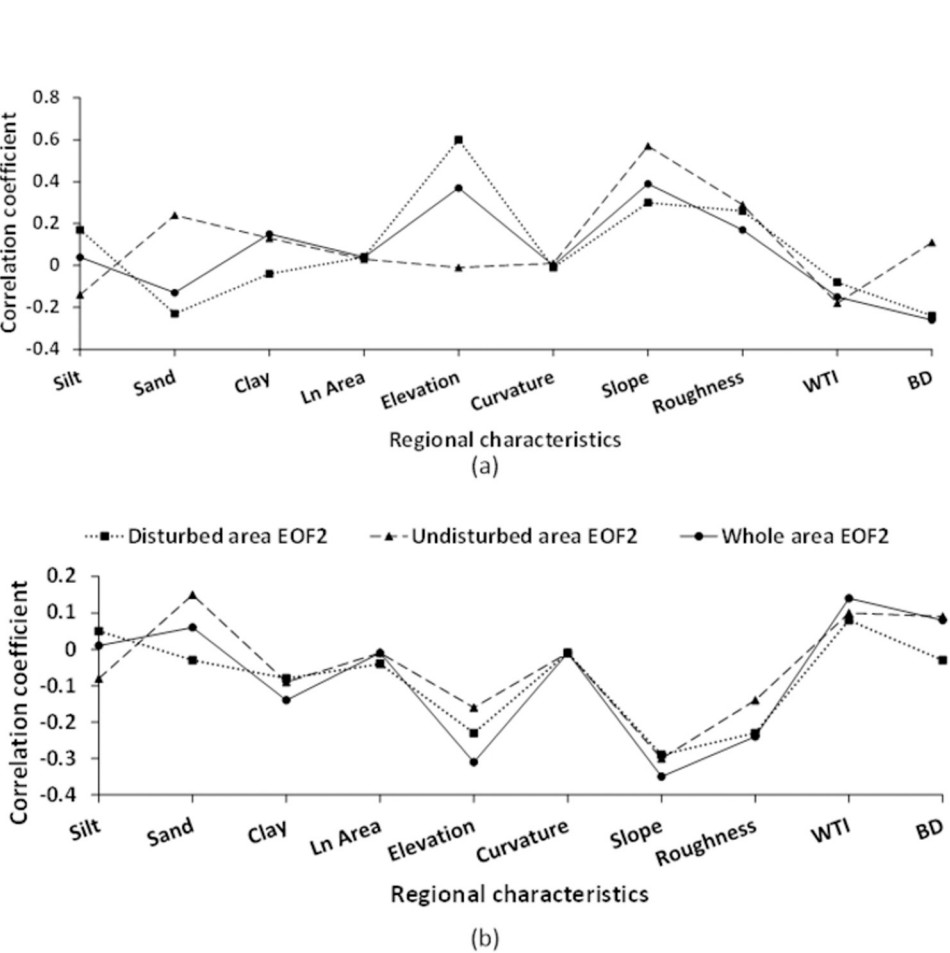

**Fig 10. Comparison of EOF modes with regional characteristics in the disturbed area, undisturbed area, and the whole area.** (a) EOF1, (b) EOF2.

the percentage of sand particles was 0.24. The correlation between bulk density and disturbed area, and bulk density and whole area were −0.24 and −0.16, respectively, while the correlation between bulk density and undisturbed area was 0.11. The correlation between the disturbed area and elevation was 0.6, and the whole area and elevation was 0.42, while the correlation between the undisturbed area and elevation was -0.01, indicating that the response direction of the disturbed area and the undisturbed area to the regional characteristics differed. Compared with the undisturbed area at a lower elevation, the disturbance area at higher elevation may have different soil texture.

Correlation analysis showed that soil texture, elevation, and slope were most strongly correlated with EOF mode. The characteristics of, natural logarithm of catchment area, and curvature were uncorrelated with EOF mode. Under drought conditions, the effects of all of the terrain features will be reduced. The reason for this result was that the terrain characteristics had an impact on soil moisture through lateral flow, and this phenomenon was not easy to observe because of the data scale and time used in this study. Future research will compare and analyze the different characteristics of soil water distribution patterns by using different resolution data, and determine the potential distribution patterns of soil water by data collected over time at various times scales.

By comparing the differences of EOF modes between the sub-regions, we try to explain how coal mining activities affect the temporal and spatial distribution of surface soil water. The correlations analysis showed that the response directions of the sub- regions to the characteristics of altitude, slope, soil texture and bulk density are obviously different. We believe that this is related to the change of the original soil structure by the land remediation measures on the damaged land.

As mentioned in the introduction, there were few studies on the temporal and spatial changes of soil surface water in mining areas of the Loess Plateau under the influence of mining subsidence. In this paper, we only focus on the correlation between soil type, topographic characteristics and soil water distribution model. However, vegetation cover also plays a very important role in the temporal and spatial distribution of surface soil water. For example, EOF1 values along both banks of the river and hillside areas were significantly different. The response directions of the two regions to time changes were opposite. On the one hand, it may be due to the horizontal redistribution of soil moisture on the date of more precipitation, on the other hand, it may be related to the planting of seasonal crops along both banks of the river, such as winter wheat. In contrast, the soil in the hillside area remains relatively dry on dry and wet days, because the soil covered by shrubs and trees will lose water due to transpiration even in wet periods. However, we lack long-term observed vegetation parameters in the study area, so there is no quantitative comparison of the impact of vegetation cover parameters on surface soil water variability in the study area. Future research will compare and analyze the soil water distribution mode by collecting long-term observed vegetation and hydrological parameters.

## 4. Conclusion

Twelve measurements of soil moisture data were used to verify the accuracy of the soil moisture data retrieved by SAR at the corresponding dates. The results showed that the soil water content estimated by the time series radar signal change detection algorithm in the study area was accurate. The average soil moisture in the surface layer of the study area varied seasonally, showing dry-wet-dry pattern. Moreover, soil moisture was significantly affected by precipitation factors, and the response to precipitation had an obvious effect. In the dry periods, soil moisture was affected by vegetation cover, the coefficient of variation of soil water content was larger, and in the wet periods, the coefficient of variation was smaller. From the perspective of spatial variation, the disturbance caused by coal mining activities did not show obvious on surface soil moisture. From the perspective of time series change, when moving from wet to dry conditions, the soil drying was faster in the disturbed area than in the undisturbed area. When moving from dry to wet conditions, soil moisture in disturbed areas was quickly increased.

The spatial and temporal characteristics of soil moisture in the study area are determined by two modes, which had different contributions in different time. The two EOFs showed that most of the observed spatial variability of soil moisture was fixed in time. The study was divided into disturbed and undisturbed area for single EOF analysis, and the results showed that the EOF mode of disturbed area is closer to that of the whole study area. Comparing the correlation between the two subregions and the whole study area and regional characteristics, the correlation values of the changes were concentrated in the characteristics of soil texture, bulk density, altitude and slope, indicating that the soil texture of the two subregions may be different at different altitudes, and may also be related to the change of the original soil structure in the disturbed area. Overall, the EOF mode of the disturbed area determined the EOF mode of the entire study area.

## Supporting information

**S1 Fig. 24 days of soil moisture data estimated based on SAR data.**
(PDF)

**S2 Fig. Data of soil texture and topographic characteristics in the study area.**
(PDF)

## Author Contributions

**Conceptualization:** Yi Cao, Rutian Bi.

**Formal analysis:** Yi Cao.

**Funding acquisition:** Hongfen Zhu, Rutian Bi.

**Investigation:** Yaodong Jin.

**Methodology:** Hongfen Zhu.

**Visualization:** Yaodong Jin.

**Writing – original draft:** Yi Cao.

**Writing – review & editing:** Yi Cao.

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
