## [Decision Letter · Decision Letter 0]

11 Sep 2021

PONE-D-21-26837Spatial and temporal characteristics of surface soil moisture in a disturbed coal mining area of Chinese Loess PlateauPLOS ONE

Dear Dr. Rutian Bi, 

Thank you for submitting your manuscript to PLOS ONE. After careful consideration, we feel that it has merit but does not fully meet PLOS ONE’s publication criteria as it currently stands. Therefore, we invite you to submit a revised version of the manuscript that addresses the points raised during the review process.

We look forward to receiving your revised manuscript.

Kind regards,

Chun Liu

Academic Editor

PLOS ONE

Journal Requirements:

When submitting your revision, we need you to address these additional requirements. 1. Please ensure that your manuscript meets PLOS ONE's style requirements, including those for file naming. The PLOS ONE style templates can be found at https://journals.plos.org/plosone/s/file?id=wjVg/PLOSOne_formatting_sample_main_body.pdf and https://journals.plos.org/plosone/s/file?id=ba62/PLOSOne_formatting_sample_title_authors_affiliations.pdf
 2. In your Data Availability statement, you have not specified where the minimal data set underlying the results described in your manuscript can be found. PLOS defines a study's minimal data set as the underlying data used to reach the conclusions drawn in the manuscript and any additional data required to replicate the reported study findings in their entirety. All PLOS journals require that the minimal data set be made fully available. For more information about our data policy, please see http://journals.plos.org/plosone/s/data-availability. "Upon re-submitting your revised manuscript, please upload your study’s minimal underlying data set as either Supporting Information files or to a stable, public repository and include the relevant URLs, DOIs, or accession numbers within your revised cover letter. For a list of acceptable repositories, please see http://journals.plos.org/plosone/s/data-availability#loc-recommended-repositories. Any potentially identifying patient information must be fully anonymized. Important: If there are ethical or legal restrictions to sharing your data publicly, please explain these restrictions in detail. Please see our guidelines for more information on what we consider unacceptable restrictions to publicly sharing data: http://journals.plos.org/plosone/s/data-availability#loc-unacceptable-data-access-restrictions. Note that it is not acceptable for the authors to be the sole named individuals responsible for ensuring data access. We will update your Data Availability statement to reflect the information you provide in your cover letter. 3. PLOS requires an ORCID iD for the corresponding author in Editorial Manager on papers submitted after December 6th, 2016. Please ensure that you have an ORCID iD and that it is validated in Editorial Manager. To do this, go to ‘Update my Information’ (in the upper left-hand corner of the main menu), and click on the Fetch/Validate link next to the ORCID field. This will take you to the ORCID site and allow you to create a new iD or authenticate a pre-existing iD in Editorial Manager. Please see the following video for instructions on linking an ORCID iD to your Editorial Manager account: https://www.youtube.com/watch?v=_xcclfuvtxQ 4. We note that Figure 1 in your submission contain [map/satellite] images which may be copyrighted. All PLOS content is published under the Creative Commons Attribution License (CC BY 4.0), which means that the manuscript, images, and Supporting Information files will be freely available online, and any third party is permitted to access, download, copy, distribute, and use these materials in any way, even commercially, with proper attribution. For these reasons, we cannot publish previously copyrighted maps or satellite images created using proprietary data, such as Google software (Google Maps, Street View, and Earth). For more information, see our copyright guidelines: http://journals.plos.org/plosone/s/licenses-and-copyright. We require you to either (1) present written permission from the copyright holder to publish these figures specifically under the CC BY 4.0 license, or (2) remove the figures from your submission: a. You may seek permission from the original copyright holder of Figure 1 to publish the content specifically under the CC BY 4.0 license.   We recommend that you contact the original copyright holder with the Content Permission Form (http://journals.plos.org/plosone/s/file?id=7c09/content-permission-form.pdf) and the following text:“I request permission for the open-access journal PLOS ONE to publish XXX under the Creative Commons Attribution License (CCAL) CC BY 4.0 (http://creativecommons.org/licenses/by/4.0/). Please be aware that this license allows unrestricted use and distribution, even commercially, by third parties. Please reply and provide explicit written permission to publish XXX under a CC BY license and complete the attached form.” Please upload the completed Content Permission Form or other proof of granted permissions as an "Other" file with your submission. In the figure caption of the copyrighted figure, please include the following text: “Reprinted from [ref] under a CC BY license, with permission from [name of publisher], original copyright [original copyright year].” b. If you are unable to obtain permission from the original copyright holder to publish these figures under the CC BY 4.0 license or if the copyright holder’s requirements are incompatible with the CC BY 4.0 license, please either i) remove the figure or ii) supply a replacement figure that complies with the CC BY 4.0 license. Please check copyright information on all replacement figures and update the figure caption with source information. If applicable, please specify in the figure caption text when a figure is similar but not identical to the original image and is therefore for illustrative purposes only.The following resources for replacing copyrighted map figures may be helpful: USGS National Map Viewer (public domain): http://viewer.nationalmap.gov/viewer/The Gateway to Astronaut Photography of Earth (public domain): http://eol.jsc.nasa.gov/sseop/clickmap/Maps at the CIA (public domain): https://www.cia.gov/library/publications/the-world-factbook/index.html and https://www.cia.gov/library/publications/cia-maps-publications/index.htmlNASA Earth Observatory (public domain): http://earthobservatory.nasa.gov/Landsat: http://landsat.visibleearth.nasa.gov/USGS EROS (Earth Resources Observatory and Science (EROS) Center) (public domain): http://eros.usgs.gov/#Natural Earth (public domain): http://www.naturalearthdata.com/

Additional Editor Comments:

Dr Rutian Bi,

I am glad to inform you that the review of this ms should be finished by two reviewers, according to these feedbacks from them, this ms could be reconsidered after a major revision. Please see specific comments. When resubmitting your manuscript, please carefully consider all issues mentioned in the reviewers' comments, outline every change made point by point, and provide suitable rebuttals for any comments not addressed.

Reviewers' comments:

Reviewer's Responses to Questions

**Comments to the Author**

1. Is the manuscript technically sound, and do the data support the conclusions?

Reviewer #1: Yes

Reviewer #2: Yes

2. Has the statistical analysis been performed appropriately and rigorously? 

Reviewer #1: No

Reviewer #2: Yes

3. Have the authors made all data underlying the findings in their manuscript fully available?

Reviewer #1: Yes

Reviewer #2: Yes

4. Is the manuscript presented in an intelligible fashion and written in standard English?

Reviewer #1: Yes

Reviewer #2: Yes

5. Review Comments to the Author

Reviewer #1: In this study, the surface soil water in 24 days was estimated using a radar signal change detection algorithm. The temporal and spatial variation characteristics of surface soil water inside and outside the disturbed area were compared and analyzed. The results showed that the soil water content estimated by the time series radar signal change detection algorithm in the study area was accurate. The average soil moisture in the surface layer of the study area varied seasonally, showing dry-wet-dry pattern. This study is very interesting. It provides a case for estimating surface soil water by using a radar signal change detection algorithm. However, it cannot be accepted for publication in current form. The detailed comments are following.

1. “These factors have left imbalances in regional water resources, so it is particularly necessary to monitor changes in soil water content in the study area”. It lacks of logic between the two sentences.

2. The deficiencies of former studies in science is not clear. What’s the contribution of this study to the science?

3. What is special of the research area? Why the soil moisture in disturbed coal mining area is important?

4. Why the surface soil moisture is important, especially in 5 cm?

5. What’s the significance of this study for application or for science?

6. Why the SAR data and CD algorithm are employed? Why they are necessary?

7. The purpose of this study should be elaborated and described clearly.

8. Figure 2. Why the relationship between backscattering coefficient difference and NDVI was expressed by using linear equation? What is the R2 of the equation? I don’t think there is obvious relationship between them.

9. Figure 3: The Chinese words should be deleted.

10. The verification and application of the results were very weak. The discussion should be improved very much.

Reviewer #2: General comments to authors:

The manuscript " Spatial and temporal characteristics of surface soil moisture in a disturbed coal mining area of Chinese Loess Plateau" shows an important issue: Temporal and Spatial Changes of Soil Moisture in Coal Mining Areas.

In this study, the author selected the coal mining area in the Changhe River Basin of the Loess Plateau as the research object. Using Synthetic Aperture Radar (Synthetic Aperture Radar) data, compared and analyzed the temporal and spatial variation characteristics of surface soil moisture within and outside the disturbance zone. And use the orthogonal function analysis method to analyze the potential time and spatial change characteristics of surface soil water in the Changhe River Basin and determine the regional distribution and change of soil moisture in the coal mining disturbance area. The selection of the research area is representative and the method is scientific, which provides a theory for ecological restoration in the mining area.

However, unfortunately the authors have not fully developed some topics of the manuscript and therefore I have decided that the manuscript needs revision. Overall, I recommend clarifying some aspects: introduction (for example, small changes are needed); of the methods in order to allow results interpretation; and results and discussion (for example, to make the results of the work more clear).

What is the practical significance of the conclusion drawn by the author, and what guiding significance and clearness does it have for other regions in China?

These are the main problems I found in the manuscript, and I hope they may help the authors when reviewing their work. Also see the PDF as several comments have also been added to it.

The detailed suggestions follow below.

Detailed comments to authors:

Abstract:

Point 1:

The research period should be clearly stated.

Point 2:

And there is a little doubt. What is the basis for selecting these 24 days, whether they are representative, and whether they can scientifically explain the changes in soil moisture in the mining area. If they can be compared in parallel periods, the conclusion of the paper will be more rigorous.

Introduction:

Point 3

The introduction should be further integrated, layer by layer, to show the innovation and practical significance of this article. In addition, the references of the article are too old to highlight the frontier research issues. you can cite other papers around the world.

Materials and Methods:

Point 4

Supplement and explain the source of the data and the selection of indicators.

Point 5

In Figure 1, "West of the Changhe River is designated as the disturbed area, and the area not affected by coal mining to the east of the Changhe River is designated as the undisturbed area", it is recommended that this part should be explained in the methodology, and Clarify the basis for the selection of disturbance area and non-disturbance area.

Point 6

In Figure 2, "Each point corresponds to a single radar signal difference……", this part should be explained in the methodology, or supplementary explanation in the form of notes.

Point 7

Regarding the methodology of "EOF Calculation", as an important part of the article, it should be explained in detail.

Results and Discussion:

Point 8

In 3.2, what does traditional analysis mean? The topic should be clear and easy to understand. Should also explain the meaning of this part, contribution to the article?

Point 9

In 3.2, line 2 of paragraph two. Is the author sure that Figure 4 can produce such a result?

Point 10

In 3.2, line 2 of paragraph three. The sources of these data are specified in the methodology.

Point 11

In 3.4, it is recommended that in the analysis of the driving factors of different models of soil moisture, the topography, soil texture and other factors selected in the study should be explained in the methodology.

Point 12

Figure 7 is recommended to be modified in accordance with the standard drawing standards, consistent with other drawings in the paper.

Point 13

The variables involved in Table 5 and Figure 9 should be supplemented in the methodology, and abbreviations should be annotated.

Point 14

For the differences of different influencing factors in different regions, conduct in-depth discussion. Where is the specific significance, the discussion should be sublimation of the result.

Point 15

The figures and tables in the manuscript are not uniform, and the fonts, sizes, and line widths in the figures are not uniform. It is recommended that all figures and tables in the manuscript are drawn uniformly in accordance with the specifications.

Conclusion:

Point 16

By displaying the results of the manuscript, it is necessary to improve the discussion part of the article. What are the practical guiding significance of these results to the non-disturbance area?

References:

Point 17

The references in the manuscript are too old to reflect the scientific frontiers of this research and the innovativeness of the problems, and do not support the results and the scientific nature of the discussion. It is recommended that the author can refer to more references around the world.

6. PLOS authors have the option to publish the peer review history of their article (what does this mean?). If published, this will include your full peer review and any attached files.

Reviewer #1: No

Reviewer #2: No

---

## [Author Response · Author response to Decision Letter 0]

20 Jan 2022

Comments from Reviewer 1：

1. “These factors have left imbalances in regional water resources, so it is particularly necessary to monitor changes in soil water content in the study area”. It lacks of logic between the two sentences.

Response: We deleted this statement and restated the purpose and significance of this study.

2. The deficiencies of former studies in science is not clear. What’s the contribution of this study to the science?

Response: The study area is located in the hilly area of the Loess Plateau with various topographic fluctuations. Traditional soil moisture monitoring sampling points are difficult to reveal the large-scale variation characteristics of the mining area, and remote sensing monitoring technology can effectively and timely obtain regional soil water information. At present, there are few studies on the temporal and spatial changes of soil surface moisture under the influence of mining subsidence.

3. What is special of the research area? Why the soil moisture in disturbed coal mining area is important?

Response: Revealing the change characteristics of soil water content and environmental driving mechanism after coal mining disturbance will have reference significance for disaster control and ecological environment restoration in mining areas, and provide reliable decision support and scientific basis for the coordinated development of hilly areas of the Loess Plateau.

4. Why the surface soil moisture is important, especially in 5 cm?

Response: Surface soil water directly affects the growth of surface vegetation. According to previous studies, the effective soil depth of soil water estimated by SAR data is 5cm. Therefore, in this study, the accuracy of SAR data is verified by the measured data of 5cm soil surface.

5. What’s the significance of this study for application or for science?

Response: In this paper, the change characteristics of soil water content in the study area were obtained to provide theoretical basis and data support for land reclamation and vegetation restoration.

6. Why the SAR data and CD algorithm are employed? Why they are necessary?

Response: The soil moisture inversion method based on change detection algorithm does not need to measure the parameters such as surface roughness and vegetation. It only needs long-time radar data and optical data to effectively remove the influence of surface roughness and vegetation

7. The purpose of this study should be elaborated and described clearly.

Response: According to 1-6 points, the introduction of this paper is restated.

8. Figure 2. Why the relationship between backscattering coefficient difference and NDVI was expressed by using linear equation? What is the R2 of the equation? I don’t think there is obvious relationship between them.

Response: We used quantile regression analysis. The relationship between backscattering coefficient difference and NDVI is linear in the statistical interval of the upper decile. Reference: Qi G , Mehrez Z , Maria E , et al. Synergetic use of Sentinel-1 and Sentinel-2 data for soil moisture mapping at 100 m resolution[J]. Sensors, 2017, 17(9):1966.

9. Figure 3: The Chinese words should be deleted.

Response: The Chinese words were deleted.

10. The verification and application of the results were very weak. The discussion should be improved very much.

Response: We added the discussion content and clarified the shortcomings of the research and the future research direction. Discussions added in 3.4：

Tiejun Wang (2017) found that the primary spatial structure of soil moisture was mostly correlated with soil texture in all the study regions, indicating the dominant role of soil in determining soil moisture spatial variability.

EOF2 was negatively correlated with altitude and slope, which also shows that the soil water distribution is more dynamic at the location with higher altitude and steeper slope. This result is consistent with previous studies. They found that the slope position is the largest contributor to the temporal variability of soil moisture. Steeper slopes are expected to promote drainage and produce higher temporal variability. Higher altitude areas may be more easily separated from stable water sources, such as rivers or regional aquifers, so the soil water change in this area is more dynamic.

By comparing the differences of EOF modes between the sub-regions, we try to explain how coal mining activities affect the temporal and spatial distribution of surface soil water. The correlations analysis showed that the response directions of the sub- regions to the characteristics of altitude, slope, soil texture and bulk density are obviously different. We believe that this is related to the change of the original soil structure by the land remediation measures on the damaged land.

As mentioned in the introduction, there were few studies on the temporal and spatial changes of soil surface water in mining areas of the Loess Plateau under the influence of mining subsidence. In this paper, we only focus on the correlation between soil type, topographic characteristics and soil water distribution model. However, vegetation cover also plays a very important role in the temporal and spatial distribution of surface soil water. For example, EOF1 values along both banks of the river and hillside areas were significantly different. The response directions of the two regions to time changes were opposite. On the one hand, it may be due to the horizontal redistribution of soil moisture on the date of more precipitation, on the other hand, it may be related to the planting of seasonal crops along both banks of the river, such as winter wheat. In contrast, the soil in the hillside area remains relatively dry on dry and wet days, because the soil covered by shrubs and trees will lose water due to transpiration even in wet periods. However, we lack long-term observed vegetation parameters in the study area, so there is no quantitative comparison of the impact of vegetation cover parameters on surface soil water variability in the study area. Future research will compare and analyze the soil water distribution mode by collecting long-term observed vegetation and hydrological parameters.

 

Comments from Reviewer 2：

Abstract:

Point 1:

The research period should be clearly stated.

Response: The research period was added: from Jan 25, 2018 to Dec 10, 2019

Point 2:

And there is a little doubt. What is the basis for selecting these 24 days, whether they are representative, and whether they can scientifically explain the changes in soil moisture in the mining area. If they can be compared in parallel periods, the conclusion of the paper will be more rigorous.

Response: Sentinel-1 radar data interval is 12 days. Sentinel-2 optical data were also used in this paper. Considering the influence of cloud cover during satellite transit on optical data, the data interval is 24 days or 36 days. The time span of this study is 2 years. Based on the data of these 2 years, we have made a methodological attempt. We will continue to conduct in-depth research. Thank you for your suggestions.

Introduction:

Point 3

The introduction should be further integrated, layer by layer, to show the innovation and practical significance of this article. In addition, the references of the article are too old to highlight the frontier research issues. you can cite other papers around the world.

Response: The research significance of this paper was restated. The latest relevant papers were reviewed.

Materials and Methods:

Point 4

Supplement and explain the source of the data and the selection of indicators.

Response: The source of the data and indicator selection were supplemented. All data can be checked in S2 Fig. and explanations were added in 2.3:

(4) Soil texture and topographic attributes

The digital elevation model (DEM) of the study area adopted ASTER GDEM. The factors such as slope, curvature, terrain roughness, natural logarithm of catchment area (Ln Area) and terrain wetness index (WTI), were extracted from the DEM of the study area. Soil texture such as silt, sand, clay and bulk density (BD) and precipitation data were obtained from the National Earth System Science Data Center (China).

Point 5

In Figure 1, "West of the Changhe River is designated as the disturbed area, and the area not affected by coal mining to the east of the Changhe River is designated as the undisturbed area", it is recommended that this part should be explained in the methodology, and Clarify the basis for the selection of disturbance area and non-disturbance area.

Response: The study area is divided into East and west areas by Changhe river. There are coal resources underground in the western region. Large area of surface damage is caused by coal mining. People have implemented large-scale land remediation on the damaged land, so we defined this region as a disturbed area. There is no coal resource underground in the eastern region, which we called the undisturbed area.

Point 6

In Figure 2, "Each point corresponds to a single radar signal difference……", this part should be explained in the methodology, or supplementary explanation in the form of notes.

Response: This sentence has been modified: Each point corresponds to the radar signal difference and NDVI in the same pixel position

Point 7

Regarding the methodology of "EOF Calculation", as an important part of the article, it should be explained in detail.

Response: A brief description of EOF calculation method was added:

For a soil moisture data set with n locations and m observations at each location, spatial anomalies of soil moisture can be computed as

〖Z_i¬¬¬¬(t)=S〗_i¬¬¬¬(t)-1/m ∑_(j=1)^m▒〖S_j¬¬¬¬(t)〗 (10)

where Zi(t) and Si(t) are soil moisture spatial anomaly and volumetric soil moisture content at location i and time t, respectively. A matrix of soil moisture spatial anomalies, Z, can be constructed as

Z=[■(Z_11&⋯&Z_1n@⋮&⋱&⋮@Z_m1&⋯&Z_mn )] (11)

Then, a covariance matrix V can be calculated as

V=1/(m-1) Z^T Z (12)

where the superscript T indicates matrix transpose. To perform EOF analysis, one needs to find eigenvectors and eigenvalues for V, which satisfy the following equation:

VE=LE (13)

where E contains eigenvectors in columns

E=[■(e_11&⋯&e_1n@⋮&⋱&⋮@e_n1&⋯&e_nn )] (14)

and L contains eigenvalues along the diagonal

L=[■(l_11&⋯&0@⋮&⋱&⋮@0&⋯&l_nn )] (15)

The above procedure rotates the original coordinate axes with each axis indicating a sampling time into a new set of orthogonal coordinate axes with each eigenvector representing a new axis. The eigenvalues explain the variance in the data along the direction of each corresponding new axis, and the portion of the explained variance (Pj) by the jth new axis in the total variance can be computed as

P_j=l_jj/(∑_(k=1)^n▒l_jj ) (16)

The eigenvectors are then arranged according to eigenvalues: the first axis explains the largest variance in the data, while each following axis explains the largest remaining variance and is orthogonal to other axes. F can be computed by projecting Z onto E

F=ZE (17)

The purpose of EOF analysis is to reduce the dimensionality of a data set, and the approach of North et al. adopted to select statistically significant EOFs. This approach assumes that an EOF is statistically significant if the lower confidence limit (e.g., 95%) of its eigenvalue is greater than the upper confidence limit of the remaining largest eigenvalue. The 95% confidence interval for the kth eigenvalue can be calculated as

〖〖CL〗_k=λ〗_k×(1±√(2/m)) (18)

where CIk is the confidence interval for the kth eigenvalue and m is the number of sampling locations.

Results and Discussion:

Point 8

In 3.2, what does traditional analysis mean? The topic should be clear and easy to understand. Should also explain the meaning of this part, contribution to the article?

Response: Compared with EOF analysis, we conducted traditional statistical analysis on the data of soil water in 24 days, such as maximum and minimum value, average value and variance. These analyses could show the periodic changes of soil water in 24 days with seasons. This part served as a supplement to EOF analysis.

Point 9

In 3.2, line 2 of paragraph two. Is the author sure that Figure 4 can produce such a result?

Response: A total of 24 periods of data were calculated in this paper, but due to space constraints, only two contrasting data were shown in this paper. We have added two periods of data to explain the relevant statements. All data can be checked in S1 Fig (Supporting information).

Point 10

In 3.2, line 2 of paragraph three. The sources of these data are specified in the methodology.

Response: The sources of data were described in materials and methods

Point 11

In 3.4, it is recommended that in the analysis of the driving factors of different models of soil moisture, the topography, soil texture and other factors selected in the study should be explained in the methodology.

Response: These data were described in materials and methods. All data can be checked in S2 Fig (Supporting information).

Point 12

Figure 7 is recommended to be modified in accordance with the standard drawing standards, consistent with other drawings in the paper.

Response: These errors have been modified

Point 13

The variables involved in Table 5 and Figure 9 should be supplemented in the methodology, and abbreviations should be annotated.

Response: These errors have been modified

Point 14

For the differences of different influencing factors in different regions, conduct in-depth discussion. Where is the specific significance, the discussion should be sublimation of the result.

Response: We added some content to the discussion section. We try to explain whether coal mining activities affect the temporal and spatial distribution of surface soil water.

Point 15

The figures and tables in the manuscript are not uniform, and the fonts, sizes, and line widths in the figures are not uniform. It is recommended that all figures and tables in the manuscript are drawn uniformly in accordance with the specifications.

Response: These errors have been modified

Conclusion:

Point 16

By displaying the results of the manuscript, it is necessary to improve the discussion part of the article. What are the practical guiding significance of these results to the non-disturbance area?

Response: We added the shortcomings of this study and the future research direction. The research results mainly focused on the impact of coal mining activities on the temporal and spatial distribution of surface soil water. The undisturbed area was only used for comparative analysis.

References:

Point 17

The references in the manuscript are too old to reflect the scientific frontiers of this research and the innovativeness of the problems, and do not support the results and the scientific nature of the discussion. It is recommended that the author can refer to more references around the world.

Response: We added the latest references related to this study

---

## [Decision Letter · Decision Letter 1]

9 Mar 2022

Spatial and temporal characteristics of surface soil moisture in a disturbed coal mining area of Chinese Loess Plateau

PONE-D-21-26837R1

Dear Dr. Bi,

We’re pleased to inform you that your manuscript has been judged scientifically suitable for publication and will be formally accepted for publication once it meets all outstanding technical requirements.

Kind regards,

Chun Liu

Academic Editor

PLOS ONE

Additional Editor Comments (optional):

Reviewers' comments:

Reviewer's Responses to Questions

**Comments to the Author**

1. If the authors have adequately addressed your comments raised in a previous round of review and you feel that this manuscript is now acceptable for publication, you may indicate that here to bypass the “Comments to the Author” section, enter your conflict of interest statement in the “Confidential to Editor” section, and submit your "Accept" recommendation.

Reviewer #1: All comments have been addressed

Reviewer #2: All comments have been addressed

2. Is the manuscript technically sound, and do the data support the conclusions?

Reviewer #1: Yes

Reviewer #2: Yes

3. Has the statistical analysis been performed appropriately and rigorously? 

Reviewer #1: Yes

Reviewer #2: Yes

4. Have the authors made all data underlying the findings in their manuscript fully available?

Reviewer #1: Yes

Reviewer #2: Yes

5. Is the manuscript presented in an intelligible fashion and written in standard English?

Reviewer #1: Yes

Reviewer #2: Yes

6. Review Comments to the Author

Reviewer #1: (No Response)

Reviewer #2: General comments to authors:

The manuscript " Spatial and temporal characteristics of surface soil moisture in a disturbed coal mining area of Chinese Loess Plateau" shows an important issue: Temporal and Spatial Changes of Soil Moisture in Coal Mining Areas.

In this study, the author selected the coal mining area in the Changhe River Basin of the Loess Plateau as the research object. Using Synthetic Aperture Radar (Synthetic Aperture Radar) data, compared and analyzed the temporal and spatial variation characteristics of surface soil moisture within and outside the disturbance zone. And use the orthogonal function analysis method to analyze the potential time and spatial change characteristics of surface soil water in the Changhe River Basin and determine the regional distribution and change of soil moisture in the coal mining disturbance area. The selection of the research area is representative and the method is scientific, which provides a theory for ecological restoration in the mining area.

In this revision, the author has done a lot of work on the improvement of the manuscript. I think the revised manuscript meets the journal publication standards.

7. PLOS authors have the option to publish the peer review history of their article (what does this mean?). If published, this will include your full peer review and any attached files.

Reviewer #1: No

Reviewer #2: No

---

## [Editor Report · Acceptance letter]

22 Mar 2022

PONE-D-21-26837R1 

Spatial and temporal characteristics of surface soil moisture in a disturbed coal mining area of Chinese Loess Plateau 

Dear Dr. Bi:

I'm pleased to inform you that your manuscript has been deemed suitable for publication in PLOS ONE. Congratulations! Your manuscript is now with our production department. 

Kind regards, 

on behalf of

Dr. Chun Liu 

Academic Editor

PLOS ONE